# Antibacterial Mechanism of Linalool against *Pseudomonas fragi*: A Transcriptomic Study

**DOI:** 10.3390/foods11142058

**Published:** 2022-07-12

**Authors:** Yuansong Li, Fei Ren, Da Chen, Haiming Chen, Wenxue Chen

**Affiliations:** 1School of Precision Instrument and Opto-Electronics Engineering, Tianjin University, 92 Weijin Road, Tianjin 300072, China; hiyuansong@163.com (Y.L.); chenda@cauc.edu.cn (D.C.); 2Hainan University-HSF/LWL Collaborative Innovation Laboratory, School of Food Sciences & Engineering, Hainan University, 58 People Road, Haikou 570228, China; rf950406@163.com (F.R.); hnchwx@163.com (W.C.); 3Tianjin Engineering Research Center of Civil Aviation Energy Environment and Green Development, Civil Aviation University of China, 2898 Jinbei Road, Tianjin 300300, China; 4Innovation Department, Hainan Province Hochung Corporation Areca-Nut Deep Processing Technology Research Institute, 45 Yuesong Road, Dingan 571299, China

**Keywords:** linalool, *Pseudomonas fragi*, transcriptomics, antibacterial mechanism

## Abstract

*Pseudomonas fragi* is the dominant spoilage bacterium that causes the deterioration of chilled meat. Our previous study showed that linalool has potent antibacterial activity against *P. fragi*, but its antibacterial mechanism is unclear. To explore the antibacterial mechanism of linalool against *P. fragi*, this study used RNA-seq technology to perform transcriptome analysis of *P. fragi* samples with or without linalool treatment (1.5 mL/L) for 2 h. The results showed that linalool treatment disrupted the extracellular lipopolysaccharide synthesis pathway in *P. fragi* and activated fatty acid metabolism and ribosomal function to compensate for cell membrane damage. The energy metabolism of *P. fragi* was severely disturbed by linalool, and multiple ATP synthases and ATP transportases were overexpressed in the cells but could not guarantee the consumption of ATP. The simultaneous overexpression of multiple ribosomal functional proteins and transporters may also place an additional burden on cells and cause them to collapse.

## 1. Introduction

“Zero Hunger” is one of the United Nations Sustainable Development Goals in 2015-2030 [1], and 2022 is an intermediate time-point in achieving this goal. Unfortunately, according to the report on the official website of the United Nations, this goal may not be achieved by 2030, which means that the issue of food security still requires great attention from all parties. Approximately 1.3 billion tons of food are destroyed or wasted for various reasons every year [2]. The loss of meat and meat products accounts for approximately 20% of this loss, and microorganisms are the main factor causing meat spoilage. *Pseudomonas fragi* has been reported to be the dominant spoilage bacterium in chilled beef, pork and seafood under aerobic storage conditions [3]. The presence of *P. fragi* can cause adverse changes in the colour, odour, viscosity and nutritional value of meat/meat products, and the most significant spoilage performance is the production of various volatile compounds [4]. Multiple reports indicated that the production of aldehydes, ketones and alcohol was associated with *P. fragi*. Papadopoulou et al. (2020) found abnormally elevated levels of ethyl caproate, ethyl caprylate and ethyl decanoate in beef contaminated with *P. fragi*, and these compounds gave beef an undesired fruity off-taste [5]. Therefore, taking effective measures to prevent *P. fragi* contamination can reduce the loss of meat.

Preservatives made from natural products showed good antibacterial properties. Linalool is a ring-free monoterpene that exists in the essential oils of many plants and demonstrates broad-spectrum antibacterial activity. According to the reports of Chen et al., He et al. and Varia et al., linalool can inhibit the physiological activities of yeast, *Staphylococcus* and *Aspergillus niger* [6,7,8]. The disruption of bacterial cell membranes by linalool resulting in the leakage of cellular contents is one of the well-established antibacterial mechanisms of linalool. In our previous studies, we found that linalool can exert a bacteriostatic effect by inhibiting the respiratory metabolism of *Listeria monocytogenes*, *Pseudomonas aeruginosa*, *Shewanella putrefaciens*, and *Pseudomonas fluorescens*, which explains the damage of linalool to bacteria from the perspective of energy metabolism [9,10,11]. While the effect of linalool on the genetic material in cells cannot be ignored, Ghosh et al. (2019) found that the combination of linalool with vitamin C and copper (LVC) promoted cell oxidation, and the free radicals generated by oxidation led to bacterial DNA damage, which achieved a bacteriostatic effect [12]. Noh et al. (2019) reported that the anticancer effect of linalool in prostate cancer cells is also related to the damage of DNA structure [13]. Slight DNA damage does not prevent microbes from multiplying but alters the expression of genes in the cell [14]. In other words, if the inhibitory concentration is not enough to kill bacterial cells, transcriptomic research is an effective means for determining the effect of antibacterial substances on bacterial cells.

The research object of transcriptome sequencing is all RNA transcripts contained in a specific cell or tissue in a certain state [15]. The sequence information of almost all transcripts in cells or tissues is comprehensively obtained by high-throughput sequencing. Transcriptome sequencing is a supportive technology (RNA-seq technology) for studying gene function and structure [16]. In recent years, the rapid development of RNA-seq technology has provided many valuable insights into the influence of natural products on bacteria-related genes. Yu et al. (2021) found that *Cinnamomum camphora* essential oil hindered the normal growth of *E. coli* by inhibiting metabolism and adhesion through RNA-seq analysis [17]. Park et al. (2019) found that the intervention of erythorbyl laurate (EL) upregulated *Staphylococcus aureus* cell wall gene expression and demonstrated that EL is a good material to act as a cell wall activator or a cell membrane targeting agent [18]. In addition, transcriptomics was used as a tool to discover new antimicrobial targets. Bacteria exposed to multiple bacteriostatic substances or antibiotics were subjected to global gene expression profiling using RNA-seq, and substances with broad targets were chosen as potential bacteriostatic agents [19,20].

In our previous work, it has been found that linalool has a good antibacterial effect on *P. fragi* [21], but the specific antibacterial mechanism is still unclear. In this study, the antibacterial mechanism of linalool against *P. fragi* was obtained by performing transcriptome analysis of *P. fragi* samples with or without linalool treatment (1.5 mL/L) for 2 h. This provides a theoretical basis for the application of linalool in food preservation.

## 2. Materials and Methods

### 2.1. Cells and Culture Conditions

*P. fragi* (ATCC 4973) was obtained from the China General Microorganism Culture Collection and Management Center. *P. fragi* was inoculated into a nutrient broth agar culture medium (NA) for passage activation. Then, *P. fragi* was inoculated in a liquid nutrient broth culture medium (NB) and cultured at 37 °C for 18 h to reach the logarithmic growth phase. The bacterial solution was centrifuged at 6000× *g* for 10 min, and the bacterial pellet was collected. The cells were washed twice with sterile normal saline and resuspended. Then, the bacterial concentration was adjusted to 10^6^~10^7^ CFU/mL by McFarland turbidimetry.

The activated *P. fragi* solution was added to NB and cultured at 37 °C for 18 h. Then, linalool was added to the treated group so that the concentration of linalool in the system was 1.5 mL/L (Minimum inhibitory concentration, MIC), which was confirmed in our previous study. An equal amount of sterile water was added to the control group. After treatment for 2 h at 37 °C, the cells were collected by centrifugation (6000 r/min, 4 °C, 10 min) and then washed 3 times with sterile phosphate-buffered saline (PBS) (0.1 mol/L, pH 7.4). Then, the clean cell pellets were quickly frozen in liquid nitrogen and stored at −80 °C until use.

### 2.2. RNA Isolation and cDNA Library Construction

Total mRNA was isolated from harvested cells using an RNA isolation kit (DP430, Qiagen, Hilden, Germany) and a Ribo-Zero rRNA Removal Kit (Bacteria, Epicentre, Madison, WI, USA), and then randomly fragmented into short fragments by adding a fragmentation buffer. The first strand of cDNA was synthesized in the Moloney Murine Leukaemia Virus (M-MuLV) reverse transcriptase system using the fragmented mRNA as a template and random oligonucleotides as primers. Subsequently, the RNA strand was degraded by RNaseH, and the second strand of cDNA was synthesized from dNTPs under the DNA polymerase I system. The second-strand cDNA was purified with a PCR purification kit (QIAQuick PCR Purification Kit, Qiagen, Hilden, Germany). The purified cDNA was end-repaired, A-tailed, and ligated with sequencing adapters. Then, AMPure XP beads (Beckman Coulter, Brea, CA, USA) were used to screen cDNAs of approximately 370–420 bp. The USER enzyme was used to degrade the second strand of cDNA containing U; then, PCR amplification was performed, and, finally, the library was obtained.

### 2.3. Transcriptome Sequencing and Quantification

The constructed library was sequenced using an Illumina NovaSeq 6000 (Illumina, San Diego, CA, USA). All data were stripped of sequences containing linkers, unknown sequences, and low-quality sequences. The clean reads were subjected to genomic mapping analysis using Bowtie2 software (Version 2.4.4, Langmead B, Washington, DC, USA). Quantitative analysis of gene expression in each sample was performed using subread software.

### 2.4. Differentially Expressed Gene (DEG) Screening and Analysis

Fragments per kilobase million (FPKM) were used as normalised values for gene expression across all samples. Then, the DEGs between the treated group and the control group were screened out according to the absolute value of log2 (fold change) > 0 along with *Padj* < 0.05. The DEGs were subjected to Gene Ontology (GO) functional enrichment analysis (http://geneontology.org/, 1 September 2021) and Kyoto Encyclopedia of Genes and Genomes (KEGG) pathway enrichment analysis (http://www.genome.jp/kegg/, 15 September 2021).

## 3. Results

### 3.1. Data Processing and Analysis

A total of 134,788,572 raw reads were collected from the Illumina sequencing platform, and 133,584,990 reads (99.11%) were considered clean reads after data filtering, sequencing error rate checks, and GC content distribution checks. The Q30 mass fraction values of samples in the two groups were both above 90%, and the GC content was above 56%. After comparing the reads of each sample with the reference genome, the proportion of all samples successfully aligned to the genome was found to be higher than 94.8%, and the proportion of multiple mapped reads did not exceed 1.8%. These results indicated that the quality of this sequencing was sufficiently high for subsequent analysis.

### 3.2. Diversity Analysis

The correlation of gene expression levels among samples is an important indicator for testing the reliability of experiments. The similarity of expression patterns between samples is proportional to the degree of correlation coefficient distance 1. As shown in Figure 1A, the correlation between the samples was greater than 0.94 in the treated group and the control group, which indicated that the parallelism of the samples was good and that the experimental data obtained were reliable. However, the correlation of the treated group vs. the control group was 0.561–0.769, indicating that linalool treatment had a significant effect on the gene expression of *P. fragi*. The PCA score plot (Figure 1B) also showed the same results: the samples in the treated group and the control group were separated, which is a manifestation of the difference in gene expression levels between the two groups.

### 3.3. DEGs

DESeq2 software was used to analyse gene expression differences, and |log2 (fold change)| > 0 and *Padj* < 0.05 were selected as the criteria for screening DEGs. Volcano plots (Figure 2A) were created to show the differential gene distribution in the treated group vs. the control group, with upregulated genes in red and downregulated genes in green. A total of 3188 DEGs were identified between the treated group and the control group, of which 1567 DEGs were downregulated and 1621 DEGs were upregulated in the treated group.

However, when the top 1% of DEGs were screened in Table 1, 4 DEGs were downregulated, and 28 DEGs were upregulated. Table 1 shows that linalool regulated 13 enzyme-related genes, including N-acetyltransferase, sulphate adenyltransferase subunit CysD, ATP synthase subunit, sulphate adenylyltransferase subunit CysN, malate dehydrogenase, thioredoxin-disulphide reductase, ferredoxin-NADP reductase, ADP-forming succinate-CoA ligase subunit beta, quinone reductase, transglycosylase, methylcrotonoyl-CoA carboxylase, serine protease, and dienelactone hydrolase. Some functional protein genes that assemble ribosomes were also regulated, such as rpsT and groL (30S ribosomal protein and chaperonin GroEL). Ribosomal proteins can affect the synthesis of various enzymes and the formation of cell membranes. In addition, the expression levels of six transporters were regulated by linalool, including the biopolymer transporter ExbD, extracellular solute-binding protein, phosphate ABC transporter substrate-binding protein PstS, ABC transporter substrate-binding protein, MotA/TolQ/ExbB proton channel family proteins, and putative porins.

The DEGs were clustered using hierarchical clustering, and the genes or samples with similar expression patterns in the heatmap (Figure 2B) were clustered together. The results showed that the responses of the samples after linalool intervention were consistent. The effect of linalool on the activity of these enzyme-related genes should be discussed in further studies.

### 3.4. GO Enrichment Analysis

GO enrichment analysis utilises a comprehensive database describing gene function, which is divided into three parts: biological process (BP), cellular component (CC), and molecular function (MF). For GO functional enrichment, *Padj* < 0.05 was used as the threshold for significant enrichment. Here, a total of 232 BP terms, 27 CC terms, and 156 MF terms were confirmed. The top 10 significant terms in each section (30 in total) were selected to form a GO bar graph (Figure 3A). The biological processes enriched for the identified DEGs included peptide biosynthesis and metabolism (GO:0043043 and GO:0006518), amide biosynthesis and metabolism (GO:0043603 and GO:0043604), and organic nitrogen synthesis and metabolism (GO: 1901566 and GO:1901564). The major cellular components enriched for the identified DEGs were the cytoplasmic fraction (GO:0044444), ribosomes (GO:0005840), intracellular organelles (GO:0043229), and protein-containing complexes (GO:0032991). The molecular functions enriched for the identified DEGs were ligase activity (GO:0016874), nucleotide-binding (GO:0000166), and oxidoreductase activity (GO:0016491). The histograms and scatterplots of significantly enriched GO terms are shown in Figure 3A,B.

### 3.5. KEGG Pathway Analysis

In the KEGG pathway analysis, a total of 80 enrichment items were identified. The top 20 significant KEGG pathways were selected to draw a histogram for display (Figure 4). Among them, a total of six significant pathways were screened under the condition of *P* < 0.05, including ribosomal pathways; fatty acid biosynthesis; oxidative phosphorylation; valine, leucine and isoleucine biosynthesis; carbon metabolism; and glycine, serine and threonine metabolism. The number of upregulated genes in these pathways was much greater than the number of downregulated genes (Table 2), which showed the trend that these pathways were promoted by linalool.

## 4. Discussion

Screening DEGs and determining gene function by transcriptome sequencing technology has the characteristics of short time consumption, low research cost, high data yield, and high data accuracy. In fact, studies of *Escherichia coli*, *L. monocytogenes*, and *S. species* based on transcriptome sequencing have been reported [22,23,24]. *P. fragi* can easily lead to the spoilage of chilled meat in aerobic storage. Linalool is an excellent natural antibacterial substance that can effectively inhibit the growth of *P. fragi* and delay the spoilage of chilled meat. Our previous research showed that cell membrane rupture and intracellular metabolic disorder appeared in *P. fragi* with linalool treatment at the minimum inhibitory concentration for 8 h [21]. In this study, we investigated the effects of linalool on cell membrane formation and the energy metabolism pathways of *P. fragi* from the perspective of molecular biology. *P. fragi* with linalool treatment was found to have a different expression of several genes from the normal state, which means that linalool can regulate the transcription of *P. fragi*.

The disruption of cell walls or cell membranes is considered to be one of the antibacterial mechanisms of linalool. Linalool can prevent the formation of the cell walls of *Listeria monocytogenes* [25] and can destroy the biofilm of various bacteria, such as *Salmonella typhimurium* [26] and *Candida albicans* [27]. Lipopolysaccharide (LPS) attached to the cell wall is a typical feature of Gram-negative bacteria such as *P. fragi* [28]. LPS is considered to be the first barrier to protect Gram-negative bacteria [29]. The destruction and inhibition of LPS synthesis is the premise for bacteriostatic substances and antibiotics to function. As shown in Table 1, the expression of gene RS05125 (regulating transglucosylase domain-containing protein) was significantly inhibited in the linalool-treated group, which hindered the transfer of core polysaccharide in LPS and further affected LPS synthesis. Phospholipids and proteins are the main components that maintain the stability of cell membranes [30]. The production of phospholipids in bacterial cell membranes relies on fatty acid metabolism pathways, and the synthesis of membrane proteins requires the expression of the encoded genes and the successful construction of ribosomes. According to the KEGG pathway analysis, the ribosomal pathways and fatty acid biosynthesis of *P. fragi* were significantly increased under the stimulation of linalool, which may be the positive feedback regulation of *P. fragi* to compensate for cell membrane damage. Specifically, RS20760 is a gene encoding an outer membrane protein that is related to the biofilm formation of *P. fragi*, and the expression level of RS20760 was significantly increased after linalool treatment, which mean that *P. fragi* supplemented the outer membrane protein. The 30S ribosomal protein S20 encoded by the rpsT gene is a key component of the ribosome, responsible for translation initiation and the association of the 30S and 50S subunits [31]. The expression of the rpsT gene is directly related to the synthesis of membrane proteins, transporters, and various enzymes in cells. Among the GO functional enrichment of DEGs, linalool significantly upregulated the expression of the S20 protein in ribosomes, which indicated that multiple proteins of bacterial cells were overexpressed.

The regulation of energy metabolism is important for maintaining cell stability. Oxidative phosphorylation is the main source of energy for *P. fragi*. The enzymes and substrates involved in the oxidative phosphorylation pathway directly affect cell function and viability [32]. In our previous research, we found that linalool leaked the respiratory chain dehydrogenase and ATP of *P. fragi*, which led to the disturbance of respiratory metabolism. The transcriptome results in this study also provide evidence in support of the above conclusion. In this study, oxidative phosphorylation was an important pathway enriched for the identified DEGs in KEGG pathway analysis. As shown in Table 1, the gene expression levels of ATP synthase, malate dehydrogenase, ferredoxin-NADP reductase, and ADP-forming succinate-CoA ligase subunit beta were significantly upregulated under the influence of linalool, which may explain how *P. fragi* cells replenish key enzymes and ATP. Sulphate adenylyltransferase is one of the key enzymes in transferring ATP from mitochondria to the cytoplasm [33], and the upregulation of genes cysD and cysN in the linalool-treated group indicated that the cytoplasm needs a sufficient ATP supply to compensate for the loss caused by ATP leakage [34]. However, this strategy did not appear to work, and there was no evidence that ATP was consumed in the cell. The hydrogen ions generated by ATP consumption were the power source of flagellar movement. Table 1 showed that the expression level of the gene RS13000 regulating flagellar basal body rod protein FlgF is decreased, which was a symptom of hydrogen ion deficiency. Together, this evidence suggested that linalool is a heavy blow to the energy metabolism of *P. fragi*.

At the same time, *P. fragi* upregulates multiple transporters, including the biopolymer transporter ExbD, the phosphate ABC transporter substrate-binding protein PstS, ABC transporter substrate-binding protein, and putative porin, to satisfy substrates required for oxidative phosphorylation supply. However, Wang et al. (2003) reported that the overexpression of transporters causes an increased burden on the cell membrane [35], which may accelerate cell collapse.

## 5. Conclusions

Overall, linalool significantly inhibited the growth of *P. fragi*. By comparing the transcript levels between the treated and control groups, it was found that linalool treatment disrupted the extracellular lipopolysaccharide synthesis pathway in *P. fragi* and activated the fatty acid metabolism and ribosomal function to compensate for cell membrane damage. The energy metabolism of *P. fragi* was severely disturbed by linalool, and multiple ATP synthases and ATP transportases were overexpressed in the cells but could not guarantee the consumption of ATP. The simultaneous overexpression of multiple ribosomal functional proteins and transporters may also place an additional burden on cells and cause them to collapse.

## Figures and Tables

**Figure 1 foods-11-02058-f001:**
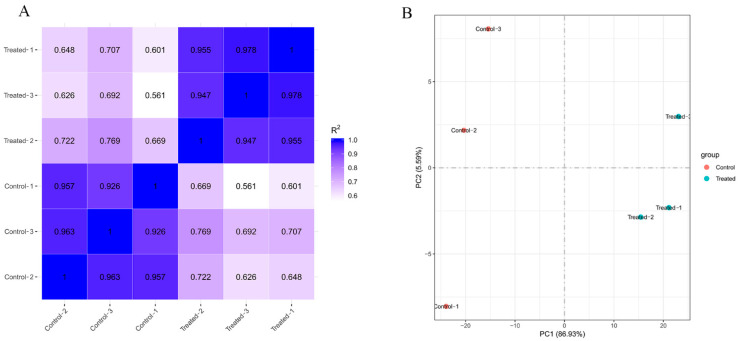
Correlation analysis of samples in the treated group and the control group. (**A**) Pearson correlation between samples. (**B**) Principal component analysis (PCA) based on FPKM.

**Figure 2 foods-11-02058-f002:**
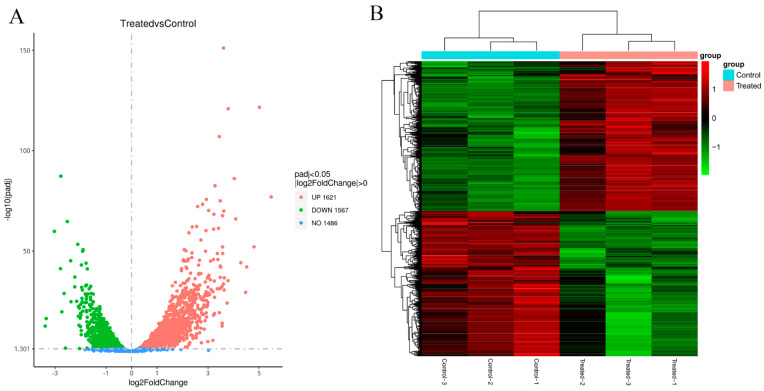
Differentially expressed gene analysis. (**A**) Volcano plot for all DEGs. (**B**) Cluster heatmap of DEGs. The columns represent samples, and the rows represent genes.

**Figure 3 foods-11-02058-f003:**
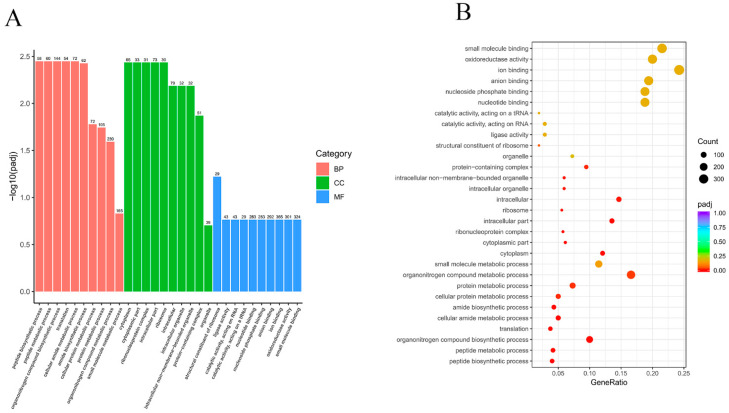
(**A**) Histogram of GO functional enrichment analysis based on DEGs. (**B**) Scatter plot of GO functional enrichment analysis based on DEGs.

**Figure 4 foods-11-02058-f004:**
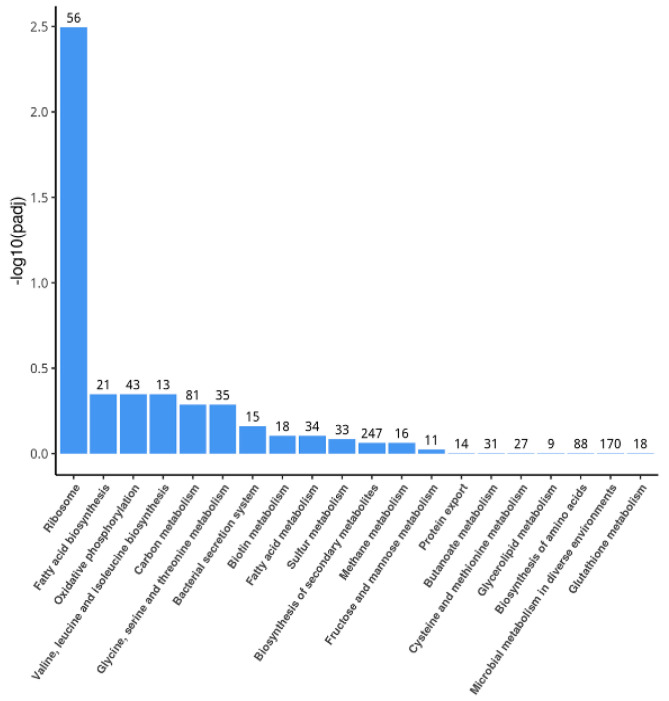
Histogram of KEGG enrichment analysis.

**Table 1 foods-11-02058-t001:** The DEGs of *P. fragi*’s responses to linalool (top 1%).

Gene Name	Description	Log2 (Fold Change)	*P*	*P adj*
RS12095	N-acetyltransferase	3.61	1.56 × 10^−155^	7.30 × 10^−152^
RS19715	biopolymer transporter ExbD	5.02	1.06 × 10^−125^	2.49 × 10^−122^
cysD	sulphate adenylyltransferase subunit CysD	3.79	8.97 × 10^−125^	1.40 × 10^−121^
RS20760	extracellular solute-binding protein	3.44	8.44 × 10^−111^	9.87 × 10^−108^
RS05965	Lrp/AsnC family transcriptional regulator	−2.77	6.23 × 10^−91^	5.83 × 10^−88^
rpsT	30S ribosomal protein S20	4.03	1.10 × 10^−89^	8.56 × 10^−87^
RS19705	TonB-dependent receptor	3.27	4.91 × 10^−86^	3.28 × 10^−83^
RS08005	phosphate ABC transporter substrate-binding protein	5.48	2.21 × 10^−80^	1.29 × 10^−77^
RS12090	1-acyl-sn-glycerol-3-phosphate acyltransferase	2.94	3.87 × 10^−79^	2.01 × 10^−76^
RS10370	SCP2 sterol-binding domain-containing protein	3.48	2.29 × 10^−78^	1.07 × 10^−75^
RS08405	F0F1 ATP synthase subunit delta	2.79	1.48 × 10^−76^	6.29 × 10^−74^
cysN	sulphate adenylyltransferase subunit CysN	2.59	1.80 × 10^−75^	7.01 × 10^−73^
RS00275	ABC transporter substrate-binding protein	3.03	1.85 × 10^−73^	6.65 × 10^−71^
groL	chaperonin GroEL	3.62	4.20 × 10^−73^	1.40 × 10^−70^
RS00755	malate dehydrogenase	3.23	1.70 × 10^−71^	5.30 × 10^−69^
RS04250	hypothetical protein	3.57	8.84 × 10^−71^	2.58 × 10^−68^
RS08110	hypothetical protein	4.09	4.11 × 10^−69^	1.13 × 10^−66^
trxB	thioredoxin-disulphide reductase	2.55	2.70 × 10^−65^	6.65 × 10^−63^
RS19700	MotA/TolQ/ExbB proton channel family protein	2.38	4.53 × 10^−65^	1.06 × 10^−62^
RS18445	YggL family protein	3.41	2.21 × 10^−64^	4.91 × 10^−62^
RS06165	hypothetical protein	3.22	6.39 × 10^−64^	1.36 × 10^−61^
RS08105	LysR family transcriptional regulator	2.92	3.53 × 10^−63^	7.18 × 10^−61^
fpr	ferredoxin-NADP reductase	2.25	4.97 × 10^−62^	9.29 × 10^−60^
sucC	ADP-forming succinate--CoA ligase subunit beta	2.72	1.91 × 10^−58^	3.43 × 10^−56^
RS13000	flagellar basal body rod protein FlgF	−2.11	2.79 × 10^−56^	4.82 × 10^−54^
RS12740	putative porin	4.80	5.39 × 10^−55^	9.00 × 10^−53^
RS22490	co-chaperone GroES	3.57	6.21 × 10^−55^	1.00 × 10^−52^
RS08580	NADPH:quinone reductase	2.07	1.34 × 10^−53^	2.09 × 10^−51^
RS05125	transglycosylase domain-containing protein	−1.90	1.58 × 10^−53^	2.38 × 10^−51^
RS06800	methylcrotonoyl-CoA carboxylase	−1.92	4.51 × 10^−53^	6.59 × 10^−51^
eco	serine protease inhibitor ecotin	2.13	5.84 × 10^−53^	8.27 × 10^−51^
RS20605	dienelactone hydrolase family protein	2.90	9.24 × 10^−53^	1.27 × 10^−50^

**Table 2 foods-11-02058-t002:** Partial results of KEGG enrichment analysis (top 10).

KEGGID	Description	*P*	*Padj*	Count	Up	Down
pfz03010	Ribosome	4.01 × 10^−5^	0.003207	56	55	1
pfz00061	Fatty acid biosynthesis	0.013608	0.450221	21	14	7
pfz00190	Oxidative phosphorylation	0.019701	0.450221	43	37	6
pfz00290	Valine, leucine and isoleucine biosynthesis	0.022511	0.450221	13	9	4
pfz01200	Carbon metabolism	0.037621	0.515821	81	59	22
pfz00260	Glycine, serine and threonine metabolism	0.038687	0.515821	35	19	16
pfz03070	Bacterial secretion system	0.060324	0.689417	15	14	1
pfz00780	Biotin metabolism	0.086515	0.784892	18	10	8
pfz01212	Fatty acid metabolism	0.0883	0.784892	34	18	16
pfz00920	Sulfur metabolism	0.102724	0.821793	33	19	14

## Data Availability

Data are contained within the article.

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
