# Peer review of "Antibacterial Mechanism of Linalool against Pseudomonas fragi: A Transcriptomic Study"

_foods, 2022, doi:10.3390/foods11142058_

Round 1

Reviewer 1 Report

The manuscript entitled ''Antibacterial mechanism of linalool against Pseudomonas fragia transcriptomic study. '' explores the antibacterial mechanism of linalool against P. fragi using RNA-seq technology.

The manuscript is concise and interesting from the medical viewpoint; however, the authors need to make some changes.

·       The authors wrote, ''In this study, transcriptomic analysis was performed on P. fragi samples with or without linalool treatment (1.5 mL/L) for 2 h to identify differentially expressed gene.''

Are two hours enough to identify expressed gene?

·       The authors should mention the source, isolation, and identification of bacteria.

·       Treatment with MIC cause inhibition of the growth of bacteria,

Why didn't the authors treat bacteria with sub-MIC to maintain bacterial viability?

·       The authors should clarify why treatment at 30 °C, not at 37°C (the optimum growth temperature of bacteria). I think this is another factor with linalool, although linalool should be the only affecting factor.

·       The authors should mention the sequence of used oligonucleotides primers.

·       Line 113 Results, not  Results and discussion.

·       Authors need to correct some grammatical mistakes throughout the manuscript.

·       Finally, I think the reference style in the manuscript is not MDPI style.

Author Response

The manuscript entitled ''Antibacterial mechanism of linalool against Pseudomonas fragi a transcriptomic study. '' explores the antibacterial mechanism of linalool against P. fragi using RNA-seq technology.

The manuscript is concise and interesting from the medical viewpoint; however, the authors need to make some changes.

  1. The authors wrote, ''In this study, transcriptomic analysis was performed on P. fragi samples with or without linalool treatment (1.5 mL/L) for 2 h to identify differentially expressed gene.'' Are two hours enough to identify expressed gene?

The protocol for treating the samples for 2 h was obtained after a series of preliminary experiments. In the preliminary experiment, we found that the differential gene expression of the 1 h treatment group was not obvious, and the cell death rate and RNA degradation rate of the 4 h-treated group were higher than 95%, but the above problems did not occur in the 2 h-treated group. Therefore, we believe that it is a scientific decision to process the samples for 2 h.

  1. The authors should mention the source, isolation, and identification of bacteria.

It has been revised according to your suggestion. (Lines 111-112 Page 2)

  1. Treatment with MIC cause inhibition of the growth of bacteria, Why didn't the authors treat bacteria with sub-MIC to maintain bacterial viability?

First, the method of this study refers to previous reports. Gao et al. (2019) also used MIC of Linalool to treat Listeria monocytogenes and performed transcriptome analysis. Secondly, the self-repair of bacterial cells when they are damaged is a life process that cannot be ignored. We believe that the use of sub-MIC will activate the expression of self-repair genes and interfere with the study of the bacteriostatic mechanism of linalool on P. fragi.

[1] Gao, Z., et al. "Anti-listeria Activities of Linalool and Its Mechanism Revealed by Comparative Transcriptome Analysis." Frontiers in Microbiology 10(2019):2947.

  1. The authors should clarify why treatment at 30 °C, not at 37°C (the optimum growth temperature of bacteria). I think this is another factor with linalool, although linalool should be the only affecting factor.

This is a writing error of the manuscript, and the culture of P. fragi was performed at 37 °C. It has been re-checked in the revised manuscript. (Lines 114, 119 and 138 Page 2-3)

  1. The authors should mention the sequence of used oligonucleotides primers.

As described in the manuscript, fragmented mRNA was used as template and random oligonucleotides as primers in the synthesis of cDNA.

The base information of the linker sequence is as follows:

5’ Adapter:

5’-AATGATACGGCGACCACCGAGATCTACAC(index)TCTTTCCCTACACGACGCTCTTCCGATCT-3’

3’ Adapter:

5’-GATCGGAAGAGCACACGTCTGAACTCCAGTCAC(index)ATCTCGTATGCCGTCTTCTGCTTG-3’

  1. Line 113 Results, not Results and discussion.

It has been revised according to your suggestion. (Line 169 Page 3)

  1. Authors need to correct some grammatical mistakes throughout the manuscript.

Following your suggestion, we have carefully proofread the revised manuscript and try our best to avoid these errors.

  1. Finally, I think the reference style in the manuscript is not MDPI style.1. Many typos and grammatical mistake are found. Writings are needed significant improvements to avoid any grammar and awkward sentences. A careful proof reading is necessary.

Following your suggestion, we have revised the reference style in the manuscript.

Reviewer 2 Report

Comment 1: Line no. 16, 34, Remove P. fragi from () from abstract and introduction.

Comment 2: Please rewrite the abstract.

Comment 3: Please check citations and references throughout the manuscript, revised as per the author's instructions.

Comment 4: Line 48: Write the first letter of the genus name in capital letter

Comment 5: Line 65-67: Please remove the sentence and write the objectives. Include this in materials and methods.

Comment 6: Line 69: Write expansion for GO and KEGG?

Comment 7: In the Introduction, please write more information about antibacterial activity and transcriptome analysis.

Comment 8: Line 81, 84: Write expansion for MIC and PBS.

Comment 9: Line 113, please change in to Results.

Comment 10: Line no. 203, when you write the second time, just write the first letter of the genus name. Please check throughout the manuscript for Scientific names.
Comment 11: Please improve the discussion section.

Comment 12: In conclusion, instead of saying general enzymes, specify the enzyme names.

Comment 13: Please check English, tyops and general mistakes, particularly references and citations.

Author Response

Comment 1: Line no.16, 34, Remove P. fragi from () from abstract and introduction.

It has been revised according to your suggestion. (Lines 16, 37 Page1)

Comment 2: Please rewrite the abstract.

Following your suggestion, we have rewritten the abstract.  (Lines 16-26 Page1)

Comment 3: Please check citations and references throughout the manuscript, revised as per the author's instructions.

Following your suggestion, we have revised the reference style in the manuscript.

Comment 4: Line 48: Write the first letter of the genus name in capital letter

It has been revised according to your suggestion. (Line 74 Page 2)

Comment 5: Line 65-67: Please remove the sentence and write the objectives. Include this in materials and methods.

We all think well of your suggestion and the objectives of this study have been written in there. (Lines 104-108 Page 2)

Comment 6: Line 69: Write expansion for GO and KEGG?

It has been revised according to your suggestion. (Lines 166-167 Page 3)

Comment 7: In the Introduction, please write more information about antibacterial activity and transcriptome analysis.

Following your suggestion, we have added more information about them. (Lines 89-103 Page 2)

Comment 8: Line 81, 84: Write expansion for MIC and PBS.

It has been revised according to your suggestion. (Line 121, Page2; Line 139 Page3)

Comment 9: Line 113, please change in to Results.

It has been revised according to your suggestion. (Line 169 Page 3)

Comment 10: Line no. 203, when you write the second time, just write the first letter of the genus name. Please check throughout the manuscript for Scientific names.

Following your suggestion, we have checked throughout the manuscript for Scientific names.

Comment 11: Please improve the discussion section.

We have revised the statements in the Discussion section based on your suggestions. (Lines 279-296 Page 8-9; Lines 317-325 Page 9)

Comment 12: In conclusion, instead of saying general enzymes, specify the enzyme names.

We all think well of your suggestion and we have revised the statements in the Conclusion. (Lines 333-346 Page 9-10)

Comment 13: Please check English, tyops and general mistakes, particularly references and citations. The list of references should be carefully checked and corrected.

We all think well of your suggestion and the list of references has been re-checked in the revised manuscript.

Reviewer 3 Report

P. fragi is one of the main spoilage bacteria that causes the deterioration of chilled meat. Linalool, a naturally occurring terpene alcohol found in many flowers, seems to be a very efficient natural occurring antibacterial agent that breaks the bacterial cell wall. This study is set to determine the antibacterial mechanism of linalool against P. fragi by transcriptomic analysis. This is a valuable study to explain the mechanism of such a significant antibacterial agent that could be useful for the preservation of meat in the long term. Analysis of P. fragi transcriptome clearly showed the genes responsible for the antibacterial effect of linalool and their mechanisms. Data obtained were well analyzed and presented in the manuscript. Therefore, I have found this study significant since genes and metabolic pathways found to be responsible for the antibacterial effect of linalool were described using a reliable modern molecular genetics tool, RNA-seq technology. 

Author Response

Thank you very much for acknowledging our work.

Round 2

Reviewer 1 Report

Authors did their best in the revision of the manuscript so I think it is in an acceptable form to me.